# Iron activates microglia and directly stimulates indoleamine-2,3-dioxygenase activity in the N171-82Q mouse model of Huntington's disease

**David W. Donley** [1,2], **Marley Realing**[3], **Jason P. Gigley**[4], **Jonathan H. Fox**[1,2]*

**1** Department of Veterinary Sciences, University of Wyoming, Laramie, WY, United States of America,
**2** Neuroscience Graduate Program, University of Wyoming, Laramie, WY, United States of America,
**3** Microbiology Undergraduate Program, University of Wyoming, Laramie, WY, United States of America,
**4** Department of Molecular Biology, University of Wyoming, Laramie, WY, United States of America

\* jfox7@uwyo.edu

**Data Availability Statement:** Raw data is available from Mendeley Data (DOI: 10.17632/z6jnx6hjcg.1).

**Funding:** Funding for this study was provided by NIH NINDS grants R01 NS079450 (Fox) and R56 NS097813 (Fox). The Wyoming NIH INBRE

## Abstract

Huntington's disease (HD) is a neurodegenerative disorder caused by a dominant CAG-repeat expansion in the huntingtin gene. Microglial activation is a key feature of HD pathology, and is present before clinical disease onset. The kynurenine pathway (KP) of tryptophan degradation is activated in HD, and is thought to contribute to disease progression. Indoleamine-2,3-dioxygenase (IDO) catalyzes the first step in this pathway; this and other pathway enzymes reside with microglia. While HD brain microglia accumulate iron, the role of iron in promoting microglial activation and KP activity is unclear. Here we utilized the neonatal iron supplementation model to investigate the relationship between iron, microglial activation and neurodegeneration in adult HD mice. We show in the N171-82Q mouse model of HD microglial morphologic changes consistent with immune activation. Neonatal iron supplementation in these mice promoted neurodegeneration and resulted in additional microglial activation in adults as determined by increased soma volume and decreased process length. We further demonstrate that iron activates IDO, both in brain lysates and purified recombinant protein ($EC_{50}$ = 1.24 nM). Brain IDO activity is increased by HD. Neonatal iron supplementation further promoted IDO activity in cerebral cortex, altered KP metabolite profiles, and promoted HD neurodegeneration as measured by brain weights and striatal volumes. Our results demonstrate that dietary iron is an important activator of microglia and the KP pathway in this HD model, and that this occurs in part through a direct effect on IDO. The findings are relevant to understanding how iron promotes neurodegeneration in HD.

## Introduction

Huntington's disease (HD) is an autosomal-dominant neurodegenerative disorder caused by a CAG-repeat expansion in exon 1 of the huntingtin gene (*HTT*) and results from expression of mutant huntingtin protein (mHTT) which misfolds in neurons and microglial cells [1,2].

program, supported by the Institutional Development Award from NIH NIGMS 2P20GM103432, provided graduate and undergraduate fellowships to DWD and MR, respectively. The funding agencies had no input on study design, data analysis, or decision to publish.

**Competing interests:** The authors have declared that no competing interests exist.

Mutant HTT results in altered transcription, axonal transport, mitochondrial function, vesicular trafficking, inflammation, and oxidative stress are some of the mechanisms implicated in HD [3]. Exactly how these pathways are related is not fully understood, but they result in progressive neuronal dysfunction and loss, eventually leading to clinical disease. Symptoms of HD encompass a hyperkinetic motor disorder, cognitive decline, and psychiatric problems [4]. Currently, there are no interventions that delay HD onset or progression.

Numerous studies have shown that iron accumulates in the brains of HD patients. Magnetic resonance imaging–based approaches have provided evidence that iron accumulation starts before clinical onset and progresses based on CAG expansion size [5–7]. Significant iron accumulation in HD patients occurs in the striatum and cerebral cortex, regions that show early degeneration, suggesting that iron accumulation may have a role in neurodegeneration [5,8]. Post-mortem studies also demonstrate increases in regional iron levels in brains from individuals with advanced HD [8–10]. However, the approaches used in these studies could not determine the cell types involved and do not reliably differentiate between the accumulation of iron in a safe form, such as ferritin-bound, and in a potentially toxic labile form. Thus it is not clear how accumulated iron contributes to HD pathogenesis.

Huntingtin protein is involved in iron homeostasis [11,12], but the mechanisms through which brain iron accumulates in HD are incompletely understood. Microglia have an important role in brain iron metabolism [13]; being phagocytic, they clear dying cells, which can result in iron uptake [14,15]. As the primary brain immune cell, microglial cells also accumulate iron in response to inflammation [16–18]. Microglial iron accumulation is present in human HD brains and is associated with an activated microglial morphology [19]. Further, HD mouse microglia have increased ferritin expression and detectable iron (III) stores [19,20]. Iron within the ferritin shell is considered a safe form suggesting that microglia in HD demonstrate a protective response to brain iron accumulation and degeneration. Microglial iron promotes oxygen radical generation that potentiates inflammation and neurotoxicity in models of Parkinson's disease [21,22]. However, in HD it is unclear if microglial iron also drives disease progression.

The kynurenine pathway (KP) of tryptophan metabolism is segregated between microglia and astrocytes, and is activated in HD [23,24]. The first step is the oxidation of tryptophan to kynurenine by the microglial enzyme indoleamine-2,3-dioxygenase (IDO). IDO is upregulated by microglial activation [25,26].and pathway activity is thought to contribute to HD progression via generation of increased amounts of neurotoxic intermediates 3-hydroxykynurenine (3-HK) and quinolinic acid [23,24,27–29]. Iron modifies KP activity by activating 3-hydroxyanthanilic acid dioxygenase (3-HAO) and promoting synthesis of neurotoxic quinolinic acid [30]. Despite this, it is unknown if iron has a key regulatory role in the KP and if it modulates gatekeeping steps such as IDO-mediated oxidation of tryptophan.

Even though HD is caused only by CAG expansion in the huntingtin gene, an estimated 60% of the non-CAG variance in age of onset is explained by environmental factors [31]. We have previously used neonatal iron supplementation (NIS) to assess the possible role of high dietary iron intake as an environmental modifier in mouse models of HD. This model is relevant, as in most populations human infants are supplemented with iron to prevent deficiency. Recommended daily allowances for iron have been developed to prevent iron deficiency, and upper limits were established to avoid short-term adverse effects [32]; however, long-term consequences of early-life NIS are not understood [33]. We have previously shown that NIS in R6/2 and YAC128 mouse models of HD exacerbates disease progression [34,35]. We have additionally shown that NIS promotes markers of mitochondrial dysfunction in R6/2 mice [36], although the cell-specific mechanisms by which NIS promotes HD in mice are unknown.

HD patients develop presymptomatic microglial activation and elevated blood cytokine levels [37–39]. Activated microglia display morphological changes that correlate with increased production of proinflammatory cytokines such as IL-1β and increased phagocytic activity [39–41]. In HD mouse models, the detrimental role of inflammation is supported by adverse effects of proinflammatory responses to lipopolysaccharide and *Toxoplasma gondii* [37,42]. Despite evidence for a role of microglial in promoting HD, the role of iron as an activator of microglial is unclear. Here we used NIS in the N171-82Q model of human HD to investigate the relationship between iron, microglial and kynurenine pathway activation, and neurodegeneration. These mice develop neurodegeneration as evidenced by loss of brain and striatal volume, and neuronal loss [43]. We report that NIS preferentially activates microglial cells in HD mice, directly activates IDO enzymatic activity, and promotes neurodegeneration. The findings demonstrate that microglial are sensitive to iron-mediated activation and that this may promote neurodegeneration in HD.

## Results

### Elevated indoleamine-2,3-dioxygenase activity in HD mouse brain is decreased by iron chelation

We measured brain IDO activity in the striatum and cortex of WT and HD mice with and without NIS. IDO activity was significantly increased in striata ($F_{(1,42)}$ = 7.77, p = 0.0075) and cortices ($F_{(1,42)}$ = 13.57, p < 0.001) of HD mice regardless of NIS (**Fig 1A and 1B**). In HD but not WT cortices, NIS significantly increased IDO activity (p = 0.016). IDO activity in HD mice that received NIS was significantly decreased by the potent iron (III) chelator deferoxamine added *ex vivo* in striata (p = 0.0067) and cortices (p = 0.0028) (**Fig 1A and 1B**). IDO has two isoforms, IDO1 and IDO2. IDO1 is highly expressed in brain microglia as well as in peripheral tissues and is a regulator of KP activity during inflammation [44–46]. The second isoform, IDO2, is more narrowly expressed, primarily in peripheral tissues, and has much lower activity as compared with IDO1 [46,47]. Therefore, we measured *Ido1* and *Ido2* transcript levels relative to beta-actin in WT and HD brains. *Ido1* was not significantly increased in the striatum ($F_{(1,20)}$ = 1.36, p = 0.2581) but was increased in the cortex of HD mice ($F_{(1,20)}$ = 5.86, p = 0.0251) relative to WT mice regardless of NIS (**Fig 1C and 1D**). We found in both the striatum ($F_{(1,20)}$ = 4.42, p = 0.0484) and cortex ($F_{(1,20)}$ = 4.82, p = 0.0402) that NIS alone significantly increased *Ido2* expression irrespective of mouse genotype (**Fig 1E and 1F**). Together, these data demonstrate that iron may have a direct role in activating enzyme activity and that *Ido* expression does not account for all the changes in enzyme activity.

### Iron directly activates IDO

To test whether iron stimulates IDO enzymatic activity, we incubated mouse brain extracts with iron (II). Iron increased brain IDO activity ($F_{(3,16)}$ = 14.00, p < 0.001) in a dose-dependent manner (**Fig 2A**). To determine if iron directly activates IDO, we used purified human IDO1. We demonstrated a dose-dependent increase in IDO activity ($F_{(5,10)}$ = 6.42, p = 0.0064) with an $EC_{50}$ = 1.24 nM for iron activation of IDO1 (**Fig 2B**). We then studied the effect of increased iron on cultured mouse microglial cells. After 2 hours of iron incubation in serum-free medium, we measured significantly increased IDO activity ($F_{(6,14)}$ = 6.04, p = 0.0027) (**Fig 2C**). To study whether this effect was a post-translational mechanism, we supplemented iron into microglial cultures in the presence of the protein translation inhibitor cycloheximide. We performed this experiment in serum-containing medium to reduce toxicity due to protein translation inhibition. Iron significantly increased IDO activity ($F_{(3,19)}$ = 14.11, p < 0.001);

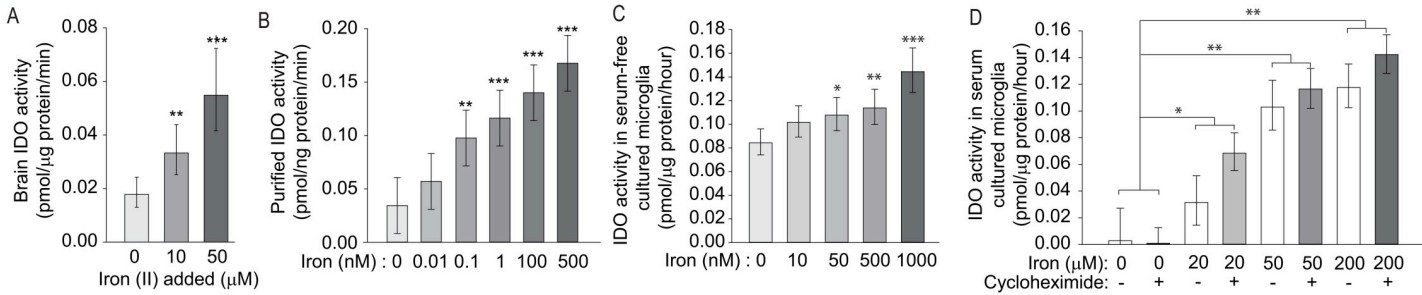

**Fig 1. Elevated IDO activity in HD mouse brains is decreased by *ex-vivo* iron chelation.** Mouse pups were supplemented with carbonyl iron from postnatal days 10–17 and then were sacrificed at 14 weeks of age. **A.** Striatal IDO activity is increased in N171-82Q HD mice and significantly decreased by *ex vivo* iron (III) chelation (50 μM deferoxamine). **B.** Cortical IDO activity is increased in iron-supplemented HD mice compared to non-supplemented HD mice and activity is decreased by *ex vivo* iron chelation. **C.** There is no difference in relative transcript levels of IDO1 in striatum. **D.** IDO1 mRNA is increased in cortex of HD mice at 14-weeks of age. **E, F.** Iron increases levels of IDO2 in striatum (**E**) and cortex (**F**) but there was no significant effect of HD on IDO2 mRNA. **A, B.** Bars represent means ± 95% CI. n = 9 wild-type and 11 HD mice in each group. *p<0.05, **p<0.01. **C, F.** Bars represent means ± SE, n = 6; *p<0.05.

**Fig 2. Nanomolar levels of iron activates IDO.** The effect of iron on IDO activity was determined in brain extracts, purified protein, and the immortalized EOC20 mouse microglial cell line. **A.** Iron (II) ammonium sulfate increases IDO activity in protein extracts from 14-week-old WT mouse brains in a dose-dependent manner. n = 5. **B.** Iron (II) activates purified IDO. n = 3. **C.** Iron(II) added to cultured microglia in serum-free medium for 2 hours increased IDO activity. n = 3. **D.** Iron(II) was added to the medium of cultured microglia for 2 hours with and without addition of cycloheximide, and IDO activity was determined. IDO activity was significantly increased by iron but there was no effect of cycloheximide. n = 6. **A, D.** Data are shown as the mean ± 95% CI. **B, C.** Data are shown as the mean ± SE. *p<0.05, **p<0.01, ***p<0.001. **A-C.** Asterisks represent comparisons to samples without iron addition.

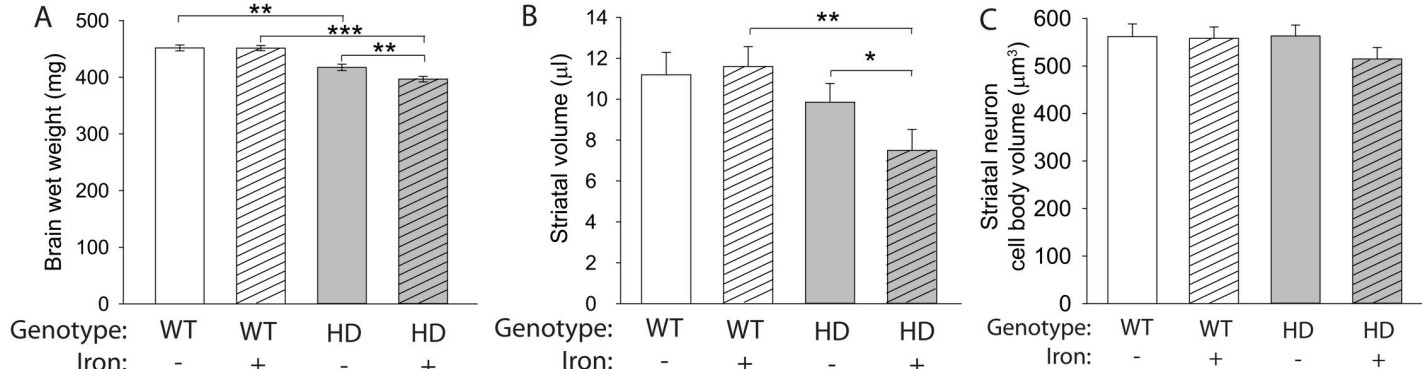

**Fig 3. Neonatal iron supplementation exacerbates HD in the N171-82Q mouse model.** Mouse pups were administered neonatal iron from postnatal days 10–17 then were sacrificed at 14 weeks of age. **A**. Total brain wet weight decreased in HD mice at 14 weeks of age. n = 15 WT control, 19 WT NIS, 13 HD control, 17 HD NIS. **B**. Striatal volume measured using the Cavalieri method was decreased in iron-supplemented HD mice compared with both control HD mice and iron-supplemented WT mice. **C**. Cell body volume of striatal neurons was not significantly different among groups. Data are shown as the mean ± SE. **p<0.01, ***p<0.001. **B, C.** n = 10 WT control, 11 WT NIS, 12 HD control, 11 HD NIS.

however, we found no difference in IDO activity with cycloheximide treatment, supporting a direct, post-translational activation of IDO by iron (**Fig 2D**).

## Neonatal iron supplementation exacerbates neurodegeneration in N171-82Q HD mice

NIS in R6/2 and YAC128 HD mouse models exacerbates neurodegeneration [34,35]. Consistent with this, both N171-82Q HD mice ($F_{(1,61)}$ = 77.70, p < 0.001) and NIS ($F_{(1,61)}$ = 4.28, p = 0.0428) significantly decreased brain weights (**Fig 3A**). There was a significant interaction ($F_{(1,61)}$ = 4.05, p = 0.0486) between HD and NIS (**Fig 3A**); iron-supplemented HD mice had significantly decreased brain weights compared with control HD mice (p = 0.0072) and iron-supplemented WT mice (p < 0.001). In addition, striatal volumes were significantly decreased in HD mice ($F_{(1,40)}$ = 11.70, p = 0.0015) ($F_{(1,40)}$ = 2.89, p = 0.0970) (**Fig 3B**). No difference in striatal volume was found comparing control HD mice with control WT mice (p = 0.3559). However, iron-supplemented HD mice had significantly decreased striatal volume compared with control HD mice (p = 0.0206) and with iron-supplemented WT mice (p = 0.0011). There was no difference among groups with respect to striatal neuron cell body volume (**Fig 3C**).

## Neonatal iron supplementation induces microglial activation in HD mice

Activated microglia demonstrate altered morphology and are present in HD [19,48]. We therefore quantified microglial morphology as an indicator of activation status. Microglia were characterized as ramified, primed, amoeboid, or reactive [48,49] (**Fig 4A**). To address whether iron supplementation and/or mouse genotype is associated with microglial activation we performed logistic regression. This allowed us to statistically test differences in the blinded, categorical assignment of microglial activation. Ramified cells are considered to be non-activated and perform a different function than activated microglia whereas the primed morphology is an intermediate functional and morphological phenotype between the non-activated ramified cells and the highly activated amoeboid and reactive microglia [50,51]. Therefore, we first compared the proportion of ramified microglia to the combined proportion of primed, amoeboid, and reactive cells. We found that iron significantly increased the probability of an activated microglial morphology. The proportion of ramified and primed microglia was then compared to highly activated amoeboid and reactive microglia to determine the probability of

a microglial cell exhibiting a fully activated morphology. We again found that iron significantly increased microglial activation and that HD mice had a higher proportion of highly activated microglia (amoeboid or reactive) when supplemented with iron (p-values in **Fig 4B**). The percentages of the four morphologies relative to total microglial cells are illustrated in **Fig 4B** for comparison purposes. To extend these findings, we also quantified morphological characteristics using a quantitative neurolucida-based approach. NIS increased cell body volumes ($F_{(1,124)}$ = 8.74, p = 0.0037), but there was no difference between iron-supplemented WT and HD microglia (**Fig 4C**). We also found a significant NIS-HD interaction with respect to the number of microglial processes ($F_{(3,40)}$ = 4.8, p = 0.006); these were significantly increased in iron-supplemented HD mice relative to control HD mice (p = 0.0024) (**Fig 4D**). In contrast, HD mice had decreased process lengths ($F_{(1,124)}$ = 6.09, p = 0.0149), which were unaffected by iron supplementation (**Fig 4E**).

**Labile iron accumulates in HD microglia.** HD microglia accumulate ferritin-bound iron, which is considered non-pathogenic but is elevated in response to increased labile iron [19,52,53]. Labile iron has the potential to drive oxidative and inflammatory processes. We used a flow cytometry to identify brain microglial cells [54] and measure cytoplasmic labile iron using the fluorescent dye calcein AM. HD mice had increased labile iron in microglial cells ($F_{(1,24)}$ = 4.67, p = 0.0409), but there was no effect of NIS on labile iron accumulation (**Fig 5A and 5B**). To assess whether iron promoted increased generalized neuroinflammation, we quantified expression of the gene encoding kynurenine monooxygenase (*Kmo*) and the chemokine CCL2 (*Ccl2*). The KMO enzyme is downstream of IDO in the KP, and its upregulation favors production of neurotoxic pathway intermediates including 3-hydroxykynurenine and quinolinic acid, which are associated with neuroinflammation [44]. The chemokine CCL2 (MCP-1), expressed by both neurons and glia, recruits microglia and peripheral monocytes to sites of inflammation in the central nervous system. Expression of CCL2 is associated with microglial activation, but is an indicator of a more generalized, sustained inflammatory response [54–56]. *Kmo* transcripts were increased both in HD mice ($F_{(1,38)}$ = 11.29, p = 0.0018) and by NIS ($F_{(1,38)}$ = 5.65, p = 0.0226) in the striatum and cortex (**Fig 6A and 6B**). In the striatum, non-supplemented HD mice showed an increase in *Kmo* transcripts compared with WT mice (p = 0.0447) and, in the cortex, iron-supplemented HD mice had increased levels of *Kmo* transcripts compared with non-supplemented HD mice (p = 0.007) and iron-supplemented WT mice (p = 0.0015). No differences in *Ccl2* expression were found in the striatum or cortex (**Fig 6C and 6D**).

## Kynurenine metabolism is increased by NIS and HD

KP metabolites provide additional markers of changes in pathway activity in response to NIS and HD. We used LC-MS to measure several metabolites simultaneously in brain. There were no changes in tryptophan levels among groups in either the cortex or striatum (**Fig 7A and 7E**). Kynurenine levels were increased by both neonatal iron ($F_{(1,80)}$ = 5.23, p = 0.0249) and the HD genotype ($F_{(1,80)}$ = 9.95, p = 0.0023) in the cortex and striatum, consistent with IDO activation (**Fig 7B and 7F**). We also found a significant effect of both neonatal iron ($F_{(1,80)}$ = 4.23, p = 0.043) and the HD genotype ($F_{(1,80)}$ = 12.7, p = 0.0006) on 3-hydroxykynurenine levels in the cortex and striatum (**Fig 7C and 7G**). Although we did not detect an effect of iron supplementation or the HD genotype alone on kynurenic acid levels in either the cortex or striatum, we did find a significant NIS-HD interaction ($F_{(1,80)}$ = 4.06, p = 0.0472) (**Fig 7D and 7H**). We also compared metabolite ratios to address the balance of kynurenine metabolism (**Fig 7I–7L**). The ratio of 3-hydroxykynurenine (3-HK) to kynurenic acid is important in determining the relative toxicity of 3-HK, with an increase in the 3-HK/kynurenic acid ratio indicating an

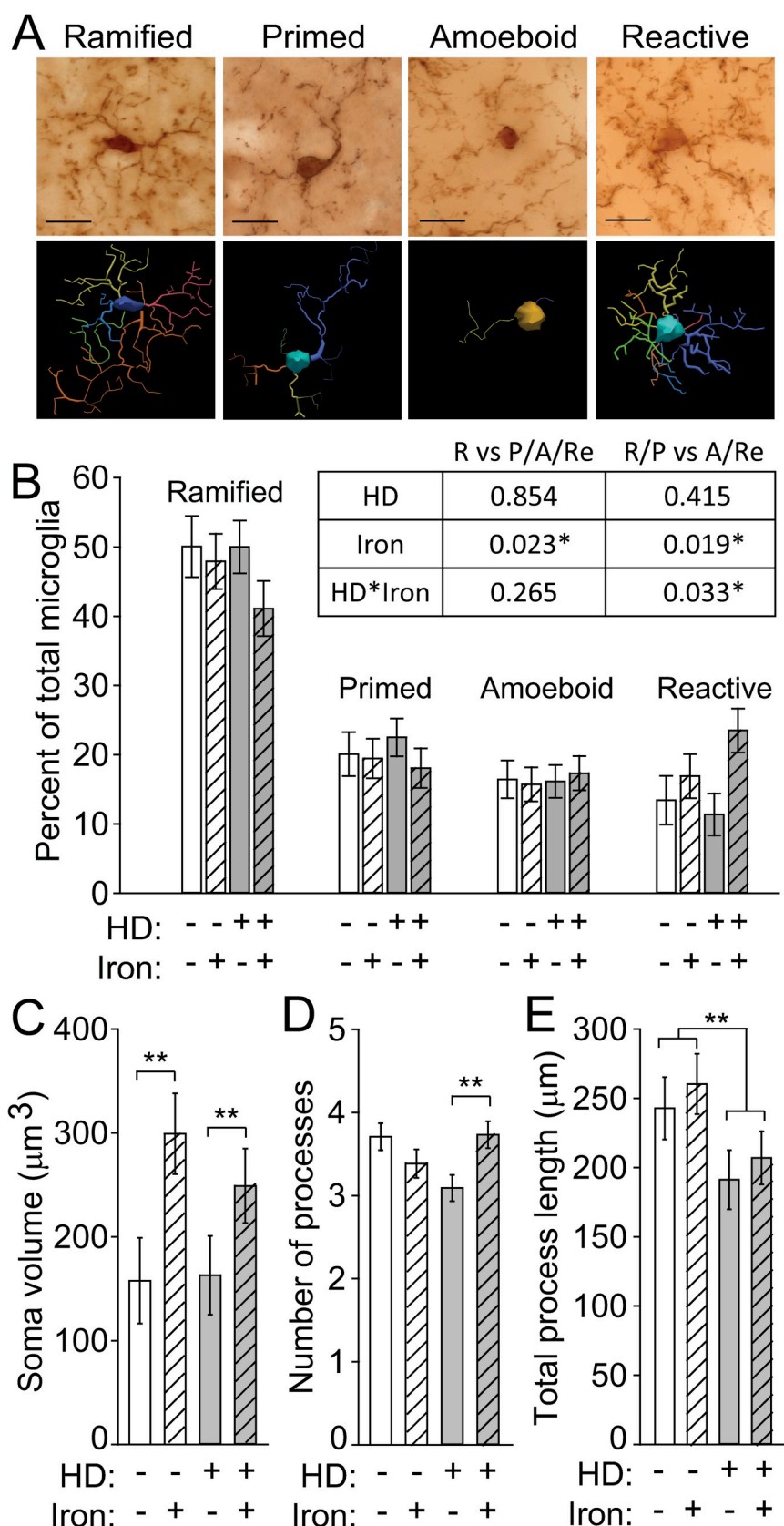

**Fig 4. Neonatal iron supplementation and HD differentially alter microglial morphology.** Microglia in brain sections from 14-week-old N171-82Q mice were labeled with anti-Iba1, and their morphology was evaluated. **A.** Representative micrographs of microglial cells labeled with anti-Iba1 were identified and characterized as having one of four morphologies, amoeboid, primed, ramified, and reactive. Neurolucida traces are shown below each image. Scale bars = 20 μm. **B.** Iron increased the frequency of reactive microglia in iron-supplemented HD mice. The table shows p-values from logistic regression analysis of microglial morphologies. Iron significantly increased the probability of activated microglial morphologies. R = ramified; P = primed; A = amoeboid; Re = reactive. Each morphology is presented as a percent of total microglial cells within each mouse. The percentage of cells with each morphology relative to all microglial cells in a given mouse are shown. **C.** The microglial soma volume was significantly increased by NIS. **D.** Iron-supplemented HD mice have a significantly increased number of processes relative to vehicle-treated HD mice. **E.** The total process length was significantly decreased in HD mice relative to WT mice but was not changed by iron supplementation. **B-F.** n = 10 WT control, 11 WT NIS, 12 HD control, 11 HD NIS. Data are shown as the mean ± SE. *p<0.05, **p<0.01.

increase in the production of neurotoxic kynurenine metabolites [27]. In both the cortex and striatum, we found that 3-HK/kynurenic acid ratios were increased in HD mice ($F_{(1,40)}$ = 7.07, p = 0.0112) but showed only a trend toward being increased by iron supplementation ($F_{(1,40)}$ = 3.09, p = 0.0864) (**Fig 7K and 7L**). In cortex, iron-supplemented HD mice had an increased 3-HK/kynurenic acid ratio compared with iron-supplemented WT mice (p = 0.0031) (**Fig 7K**).

## Discussion

Microglia are key immune cells in the brain and adapt to their environment by transitioning through a range of activation states. Neuroinflammation promotes microglial iron uptake [17] and microglial iron is increased in post-mortem human HD brain [19]. However, the effect of brain iron stress on microglial activation status in HD is less clear. By using the NIS model of promotion in a genetic mouse HD model we demonstrate that iron challenge promotes morphologic and biochemical evidence of microglial activation which is linked to neurodegeneration. The findings support a role of microglia in mediating the effects of iron on disease promotion in HD.

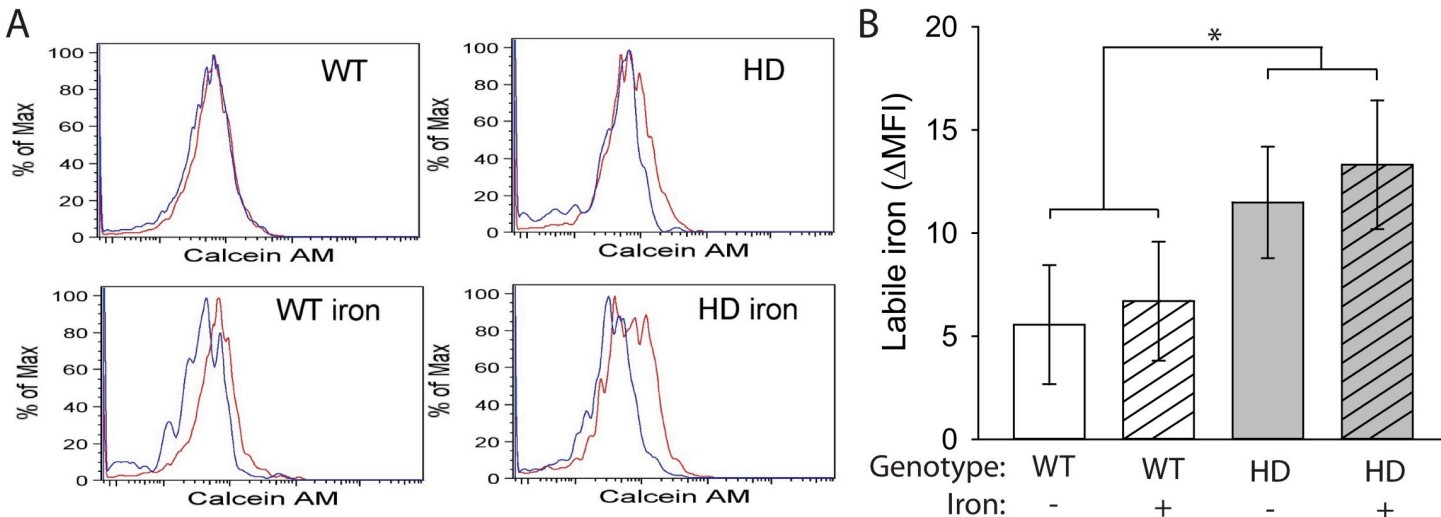

**Fig 5. Labile iron accumulates in HD microglia.** Microglia, defined as CD11b[+]CD45[+]CX3CR1[+] cells, were extracted from mouse brains at 14 weeks of age and labeled with calcein AM. **A**. Representative histograms of calcein AM fluorescence in brain microglia incubated with (red) or without (blue) deferiprone. **B**. Labile iron, defined as the difference in mean fluorescence intensity (MFI) between cells incubated with and without deferiprone, was significantly increased in HD microglia. n = 6–8. Data are shown as the mean ± SE. *p<0.05.

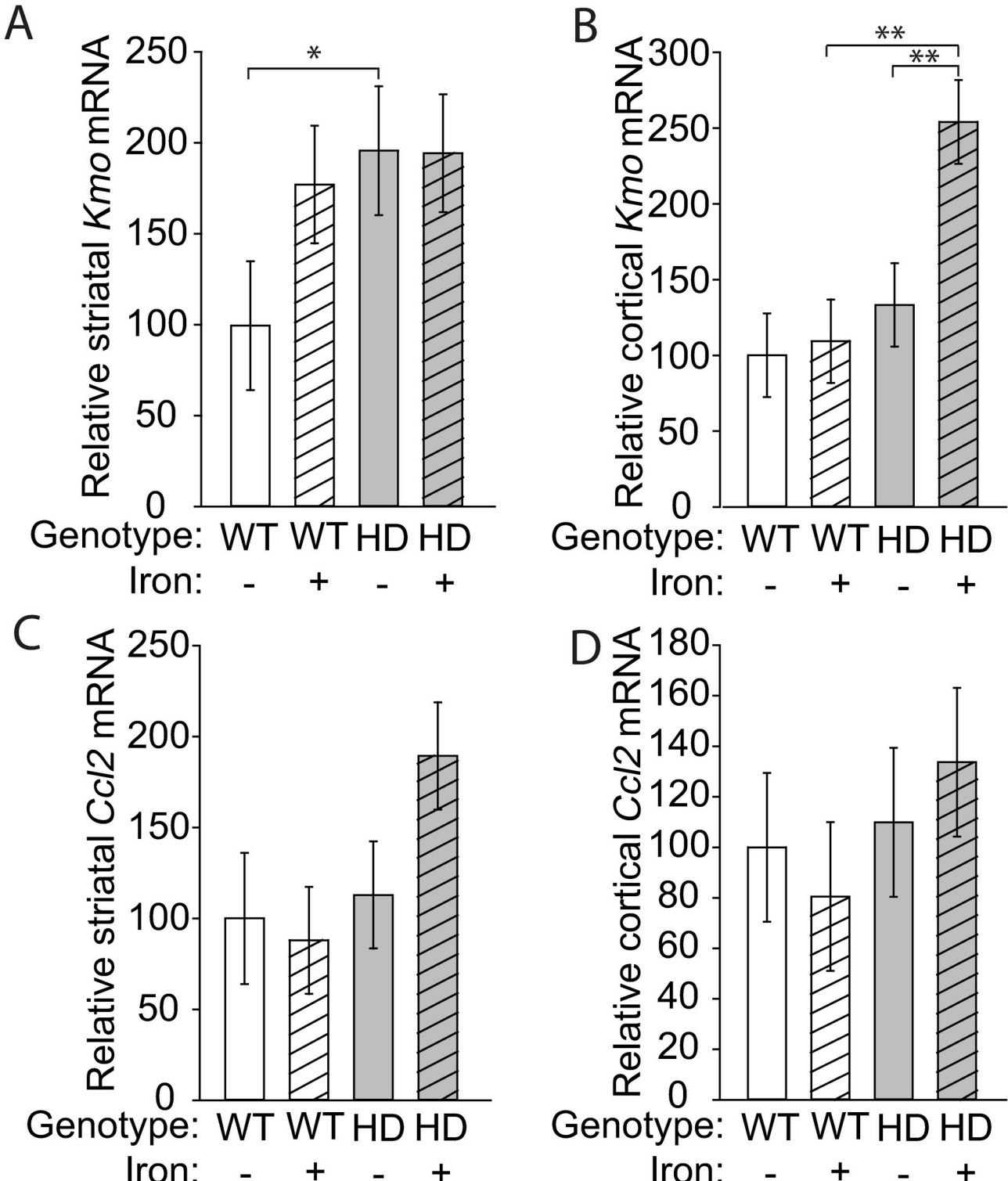

**Fig 6. *Kmo* transcripts are increased by HD and NIS.** Gene expression was determined at 14 weeks of age. **A.** *Kmo* expression was increased in the mouse HD striatum relative to WT but NIS had not effect. **B.** *Kmo* expression in iron-supplemented HD mice was significantly increased in the cortex. **C-D.** *Ccl2* expression was unchanged in the striatum (**C**) and cortex (**D**). n = 6.Data are shown as the mean ± SE. *p<0.05.

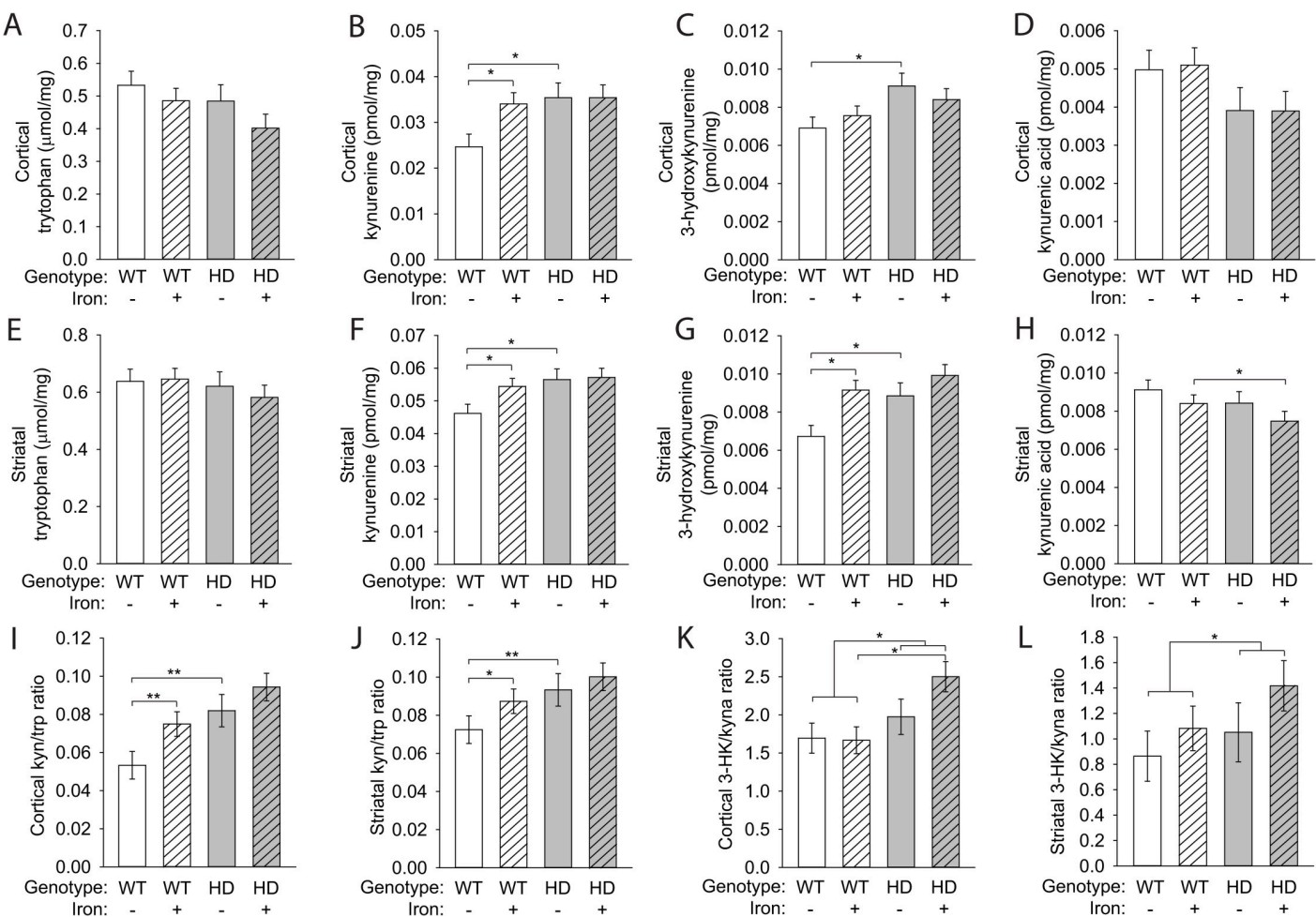

**Fig 7. Brain kynurenine metabolism is altered by the HD genotype and NIS.** Mouse pups were supplemented with iron from postnatal days 10–17. KP metabolite levels were measured in the cortex (**A-D**) and striatum (**E-H**) at 14 weeks of age and are presented per milligram of soluble brain protein. **A.** Cortical tryptophan levels were unchanged. **B.** Cortical kynurenine was increased in iron-supplemented WT mice and in HD mice relative to control WT mice. **C.** HD mice had elevated 3-HK in the cortex. **D.** Kynurenic acid levels in the cortex were unchanged as a result of iron supplementation or the HD genotype. **E.** Striatal tryptophan levels were unchanged. **F.** Kynurenine was increased in the striatum of iron-supplemented WT mice and in HD mice relative to control WT mice. **G.** Striatal 3-HK was elevated by iron supplementation in WT mice and by the HD genotype **H.** Iron-supplemented HD mice had decreased striatal kynurenic acid compared with iron-supplemented WT mice. **I.** Cortical kynurenine/tryptophan (kyn/trp) ratios were increased in iron-supplemented WT and in HD mice. **J.** The kyn/trp ratio in the striatum was increased by iron supplementation and in HD mice relative to control WT mice. **K.** Iron-supplemented HD mice had increased 3-hydroxykynurenine/kynurenic acid (3-HK/kyna) ratios in the cortex compared with iron-supplemented WT mice. **L.** Striatal 3-HK/kyna ratios were increased by the HD only in the presence of iron supplementation. **A-L.** Data are shown as the mean ± SE. n = 10.

## Neonatal iron supplementation activates microglial cells and promotes neurodegeneration in HD mice

Neonatal iron supplementation (NIS) is relevant to understanding the mechanisms through which iron promotes HD neurodegeneration because it models an iron-fortified diet that a human infant may receive, and as there is no detectable short-term toxicity [33]. Here we used the N171-82Q mouse model of HD for the first time to study the effects of NIS on microglial activation and neurodegeneration. We demonstrate that NIS potentiates neurodegeneration in these mice which is consistent with our finding in two other HD mouse models [34,35]. Microglial activation results in a graded transition of morphologic changes from ramified cells, which have many processes and small somas, to phagocytic cells, with their large somas

and short processes [49]. To assess the role of a microglial pathway in promotion of neurodegeneration we used blinded categorical and continuous methods in parallel to assess microglial morphology. We demonstrate that N171-82Q mice have morphologic evidence of microglial activation as has been reported in the R6/2 model and human HD [19]. Importantly, activation status was further enhanced by NIS. The categorical assignment of brain microglia (ramified, primed, amoeboid, reactive) is the sum of several changes including increased soma size, a change in the number of processes, and a decrease in process length. NIS in WT mice led to increased soma volume and did not change the categorical assignment of microglia. However, iron-supplemented HD mice displayed increased soma volume and increased numbers of processes (resulting from NIS), and had increased microglial activation according to categorical assessment. These data indicate that microglial activation in iron-supplemented HD mice results from a synergism between iron and HD.

Cellular iron status is sensed by the central regulators of iron metabolism, iron-regulatory proteins 1 and 2 (IRP1/IRP2) [57]. Reports of decreased IRP1/IRP2 combined with increased microglial ferritin are consistent with iron stress in HD mice [19,20]. Iron stress is thought to result from labile iron which can generate hydroxyl radicals via redox chemistry, and modify iron-sensitive enzymes [58]. By isolating brain microglial cells and using flow cytometry, we have demonstrated for the first time that HD microglia have increased labile iron which supports the presence of iron stress in these cells. The calcein AM ester used to quantify labile iron is primarily a cytoplasmic iron marker because the esterases that trap the calcein after cleavage of the acetoxy-methyl group are predominantly cytoplasmic [59]. The failure to identify an effect of NIS on labile microglial iron could therefore also be explained by location of the iron pool, or possibly insufficient experimental power.

## Labile iron directly activates IDO

The kynurenine pathway of tryptophan degradation is involved in the progression of HD and other brain disorders [44]. Activation of this pathway occurs early in human HD [23,24]. In addition, inhibition provides protection in HD model systems [29,60]. IDO is a cytoplasmic enzyme that is expressed in microglia and has an important role in regulating pathway activity. We have previously shown that IDO activity is increased in N171-82Q HD mice and that additional activation occurs in the presence of an infectious neuroinflammatory stimulus [42]. Consistent with a role in stimulating KP activity, we show that iron activated IDO in cell culture. *Ido* expression is primarily driven by pro-inflammatory signals such as lipopolysaccharide acting through the NLRP3 inflammasome [61], and IFNγ [25]. As iron can be pro-inflammatory we assessed the effect of NIS on *Ido* expression. Consistent with our prior study [42], we found increased cortical *Ido1* in non-supplemented N171-82Q HD mice only. We additionally demonstrate using recombinant IDO, cultured microglia and brain homogenates that iron results in direct activation of IDO (**Fig 2**)**.** The labile iron pool has been reported to be as high as 100–300 nM, but fluctuates based on cell type and environment [53,62,63]. We found that iron activates recombinant IDO with an $EC_{50}$ of 1.24 nM which is in the physiological range for iron activation to occur in cells. Furthermore, supporting a direct effect of iron on IDO activity *in vivo* activation by NIS in HD mice was decreased by iron chelation *ex vivo* (**Fig 1**). Together, these data are consistent with a model whereby IDO activity in HD microglia results from basal gene upregulation by inflammatory signaling and direct activation by labile iron. IDO is inhibited at the post-translational level by nitric oxide [64,65]. However, to our knowledge, this is the first report of a post-translational activator of IDO. Previous structural analyses have identified allosteric sites on IDO. Based on our data showing that IDO activity is decreased by iron chelation in brain tissue, it is possible that iron acts at an allosteric

site to increase activity [66,67]. One prevailing hypothesis of how IDO works is that tryptophan oxidation is accomplished through stepwise addition of oxygen molecules followed by a ring-break [68,69]. Therefore, it is also possible that free iron around the active site contributes to the ferryl oxygen and/or the tryptophan-epoxide intermediates during the nucleophilic addition of the second oxygen. Supporting diverse effects of iron on the KP, another study demonstrated that 3-hydroxyanthranilic acid dioxygenase, a downstream KP enzyme, is also activated by iron [30].

### Neonatal iron supplementation activates kynurenine metabolism

To further investigate KP pathway responses in HD and NIS we quantified *Kmo* transcript and four key KP pathway metabolite levels. Despite demonstrating increased IDO activity in HD mice (**Fig 1**) similar increases in kynurenine, the product of this reaction, were not found suggesting increased activity of the downstream pathway. In one branch of the KP. KMO catalyzes the conversion of kynurenine to 3-HK in the first step of quinolinic acid synthesis, a pathway that promotes neurotoxicity [44]. We assessed this step by analysis of *Kmo* transcript and 3-HK levels; elevation of both in striata of NIS wild-type mice is consistent with activation of this step. However, *Kmo* and 3HK levels in cortex were discordant across experimental groups demonstrating a complex relationship between transcript and 3HK. CCL2 is expressed in neurons and glia and has a role in the activation and recruitment of immune cells, including microglia [55,56]. The failure to find significant changes in *Ccl2* transcript levels suggests that there is not a generalized inflammatory response in HD/NIS mice, but a more specific microglial response involving the KP. In the second and neuroprotective branch of the KP, kynurenic acid is synthesized from kynurenine [44]. 3HK/kynurenine ratios, indicating toxic/protective pathway activities were increased by HD in striata and cortices with possible trends to additional increases with NIS. These findings indicate that NIS has differential effects in wild-type versus HD striata and cortices.

Taken together this study provides data supporting a causal relationship between NIS in the N171-82Q mouse model of HD, microglial activation, and neurodegeneration. NIS activated microglial differentially in wild-type and HD mice. However, NIS only promoted neurodegeneration in HD mice demonstrating specifically vulnerability to the effects of NIS in this nutritionally-relevant model. Individuals with HD have a pro-inflammatory macrophage phenotype and elevated peripheral TNFα [37,70]. The finding of IDO activation by iron is novel and relevant to not only neurodegenerative processes but also cancer where IDO is implicated in immune regulation [71]. Additional studies could focus on the mechanism of iron activation of IDO, and how NIS alters the microglial inflammatory phenotype in HD.

## Materials and methods

### Materials

All reagents were from Sigma unless otherwise stated. Validation of antibody was performed by the manufacturer. There are no restrictions on availability.

### Mouse husbandry and breeding

N171-82Q Huntington's disease mice (Jackson Labs, Bar Harbor, ME; strain B6C3-Tg (HD82Gln)81Gschi/J) were maintained as described [42]. Briefly, mice were group housed with ad-libitum access to food and water and were maintained in a 12-hour light/dark cycle. Sentinel mice were evaluated every 6 months by comprehensive serology panel screening for murine infectious diseases at Charles River Laboratories and were free of all tested diseases.

Mice received daily welfare checks. At the designated endpoint, mice were humanely sacrificed using intraperitoneal injection of B-euthanasia solution in the home cage. This study and all procedures used were approved by the University of Wyoming Institutional Animal Care and Use Committee (UW-IACUC) (protocol number 20150813JF00190-01) in accordance with the National Institutes of Health guidelines.

## Experimental design

N171-82Q HD mice were used for all experiments. The N171-82Q mouse is a transgenic model of HD that expresses the N-terminal 171 amino acids of human huntingtin protein with 82 poly-glutamine repeats under the control of the prion protein promotor [72]. When both males and females were used for experiments, the sex of the mice was included in the statistical analysis. Mouse pups received carbonyl iron supplementation or vehicle control by oral gavage from postnatal day 10–17 as described [34,35]. HD and WT littermate mice were assigned a unique 3-digit identifiers and systematically assigned to experimental groups at weaning to balance their ages and sex and to minimize effects of litter of origin. Experiments were set up with a 2 × 2 factorial design with genotype and iron treatment as the factors. Endpoints and outcomes were pre-determined. All researchers were blinded to the genotype and treatment group at weaning and remained blinded until data analysis. All experimental animals are accounted for in the data presented with the exception of three spontaneous mouse deaths. One iron-supplemented WT mouse (at 7 weeks of age) and two control HD mice (at 11 and 13 weeks of age) died during the reported studies. This study was not pre-registered.

## Brain indoleamine-2,3-dioxygenase activity

Dissected brain regions were snap frozen on dry ice and stored at −70°C until analysis. Samples were prepared and IDO activity was determined by HPLC-MS/MS as described [42]. Additionally, a separate aliquot was analyzed the same way but with addition of deferoxamine to the incubation buffer.

## Brain gene expression by real-time quantitative PCR analysis

Total RNA was extracted from the cerebral cortex and striatum using phenol/chloroform and then the cDNA was prepared, and *Ido* expression was quantified as described [42]. *Kmo* and *Ccl2* expression was analyzed using Applied Biosystems Taqman gene expression primer/probe combinations Mm1321343_m1 (*Kmo*) and Mm00441242_m1 (*Ccl2*). Expression was normalized to beta-actin using the Applied Biosystems Taqman gene expression primer/probe combination Mm00607939_s1. Gene expression was determined using 20 ng of cDNA per reaction.

## Purified indoleamine-2,3-dioxygenase activity

Recombinant human IDO (Accession #P14902) was obtained from R&D Systems (#6030-AO-010) and frozen at −80°C until use. The incubation buffer was prepared by mixing 800 μM tryptophan, 40 mM ascorbic acid, 9000 U/ml catalase, and 40 μM methylene blue in HBSS (pH = 6.5) with 80 mM ascorbic acid in 0.405 M Tris (pH = 8.0) in equal parts and pre-warmed to 37°C. One hundred ng/ml purified protein (100 ng/ml) in HBSS was mixed in equal parts with the incubation buffer and incubated at 37°C for 10 minutes. The reaction was stopped by the addition of 20% (v/v) 5 M acetic acid and incubated at 50°C for 10 minutes. Samples were centrifuged at 12,000×g, 4°C for 10 minutes, and the supernatants were filtered

through a 0.2-μm filter. Kynurenine was measured in the filtered supernatant as described [42].

## Indoleamine-2,3-dioxygenase activity in cultured cells

EOC20 cells, an immortalized microglial cell line (ATCC CTRL-2469) were used. Cells were cultured according to the manufacturer's instructions and were verified to be mycoplasma free just prior to experiments. For IDO measurements, cells were washed twice with prewarmed HBSS and then incubated with IDO incubation buffer as described [42] for 4 hours. The kynurenine concentration was quantified in the cell supernatant using the method described above. Cells were lysed, and protein levels quantified using a Bradford assay. IDO activity was defined as picomoles of kynurenine produced per microgram of protein per hour.

## Brain kynurenine metabolite quantification

Dissected brain tissue was snap frozen on dry ice and stored at −80˚C until analysis. Brain regions were prepared according to the protocol for IDO activity in brain tissue. Samples were prepared by mixing 450 μg brain protein 1:2 (v/v) with methanol containing 2% acetic acid and then centrifuging the mixture at 12,000×g for 10 minutes at 4˚C. Supernatants were filtered through a Phenomenex Phree Phospholipid extraction column and a 0.2 μm filter. The samples were then run on a Waters Acquity UPLC-MS/MS with a methanol/2% acetic acid mobile phase gradient on a $2 \times 100-$mm Waters BEH $C^{18}$ column. The sample (5 μl) was injected with a total flow rate of 0.3 ml/min and a total run time of 4.5 minutes. Analytes were verified using two M+H parent-daughter transitions. Tryptophan parent m/z = 205.13 and daughter m/z = 118.03 and 146.07; kynurenic acid parent m/z = 190.07 and daughter m/z = 88.96 and 116.03; kynurenine parent m/z = 209.04 and daughter m/z = 94.02 and 146.03; 3-HK parent m/z = 224.98 and daughter m/z = 110.04 and 162.05. Concentrations were quantified using QuanLynx (Waters) software from serial dilutions of standards.

## Mouse neuropathology

Mice designated for stereological analysis were perfused with 4% paraformaldehyde and treated as described [34,35]. Briefly, fixed brains were sectioned at 40 μm on a freezing microtome, and serial sections were collected in 12-well plates so that one well contained every 12[th] brain section. Sections from one well were mounted and thionin stained. The Cavalieri method and nucleator methods in StereoInvestigator (MicroBrightField) software were used to determine striatal volume and neuronal cell body volume, respectively.

## Microglial immunohistochemistry

Microglial morphology was characterized in the striatum at the level of the anterior commissure. Striatal sections were mounted, and microglia were immunolabeled with anti-Iba1 (RRID: AB_2636859, Abcam ab178846). Brain sections were permeabilized for 15 minutes in methanol containing 3% hydrogen peroxide and then blocked with 1× TBE, 0.05% (v/v) Tween-20, and 2% (v/v) normal serum. Sections were incubated for 3 days in primary antibody (1:1000) with appropriate controls in blocking buffer at 4˚C. They were then washed and incubated for 24 hours with biotinylated secondary antibody prior to washing, labeling with streptavidin-HRP conjugate (Vector Labs, RRID:AB_2336827), and detection using 3,3'-diaminobenzidine per the manufacturer's instructions. Reactions were quenched with water and then washed twice in 1× TBE prior to coverslipping with fluoromount G (Southern Biotech).

## Microglial morphology characterization

Microglial cells were defined as Iba1-positive cells in brain sections. Two 50 μm image stacks were collected from the dorsal and medial portion of the striatum at the level of the anterior commissure. From these images, two independent approaches were used to quantify the microglial morphology. First, cells were classified as ramified, primed, reactive, or amoeboid as described [49]. Second, quantitative morphology parameters were determined by uploading images to Neurolucida software (MicroBrightField) and each cell was traced throughout the stack. All cells that were fully in the image stack were evaluated. Ten microglial cells or more were traced per brain.

## Microglial labile iron analysis

We used flow cytometry to identify microglia purified from forebrains in combination with calcein AM detection of labile iron. In brief, mice were perfused for two minutes with cold, heparinized saline. Forebrains were dissected out and placed in cold HBSS and then diced, digested, and homogenized to isolate brain microglia as described [73,74]. Cells from each animal were plated at $1 \times 10^6$ cells/ml in four separate wells. Cells were incubated with primary antibody conjugates for 30 minutes. Microglial cells were labeled using CD11b, CD45, and CX3CR1 (Biolegend, RRID: AB_312791, AB_2563538, AB_2565700 respectively) and analyzed by flow cytometry. Microglial markers including anti-Iba1 and CD68 were not used at the risk of losing iron from the cell due to the permeabilization required for intracellular staining. Previously, calcein AM was used to characterize the labile iron pool in cells by comparing calcein AM fluorescence with and without an iron chelator added to cells [75,76]. Two wells from each brain were incubated with 10 μM deferiprone, a membrane permeable iron chelator, in HBSS for 30 minutes at room temperature, and two wells received buffer at the same time. Fixable Near-IR Live/Dead Stain (ThermoFisher Scientific) was used to exclude dead cells. To detect iron, the cells were also incubated with 20 μM calcein AM for 15 minutes. Cells were washed thoroughly, and flow cytometry data were collected using a Guava easyCyte HT flow cytometer (Millipore). Data were analyzed using FlowJo v9.3 software. The cell gating and compensation strategy was determined from no-stain, single-color, and fluorescence-minus-one controls for each fluorescence channel. The mean fluorescence intensity was determined for each sample. Labile iron in microglia was defined as the difference in the mean fluorescence intensity for cells from the same mouse that were labeled with calcein AM with and without deferiprone treatment, similar to that described previously [75,76].

## Statistical analyses

Data were analyzed using SAS software version 9.2 (Cary, NC). Generalized linear modeling was used for single-time point analyses, and mixed modeling was used for repeated measures and matched data. Assumptions of normality and equal variance were verified. All main effects and interactions were investigated in the initial statistical model, and then we performed pre-planned pairwise comparisons. Non-normally distributed data were log transformed for analysis and are presented as the mean ± 95% CIs. Logistic regression was used to analyze the categorical data on microglial morphology and to determine interaction effects between treatment and genotype. Logistic regression yielded an inference on the odds of microglial activation based on an activated microglial morphology. This statistical method is powerful against non-linear effects and for data, such as microglial morphology, where the interval between variables is not constant. Sample sizes were determined based on previous experience using each outcome. Data points represent the average of technical duplicates unless otherwise indicated. Differences with $p < 0.05$ were considered significant.

## Acknowledgments

The authors would like to thank Dr. Kenneth Gerow, University of Wyoming, for consulting on statistical analysis of microglial morphologic data.

## Author Contributions

**Conceptualization:** David W. Donley, Jason P. Gigley, Jonathan H. Fox.

**Data curation:** Marley Realing.

**Formal analysis:** David W. Donley.

**Funding acquisition:** Jason P. Gigley, Jonathan H. Fox.

**Investigation:** David W. Donley, Jason P. Gigley.

**Methodology:** David W. Donley, Marley Realing.

**Project administration:** Jason P. Gigley, Jonathan H. Fox.

**Supervision:** Jason P. Gigley, Jonathan H. Fox.

**Writing – original draft:** David W. Donley.

**Writing – review & editing:** Jason P. Gigley, Jonathan H. Fox.

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
