## [Decision Letter · Decision Letter 0]

19 Mar 2021

PONE-D-21-02271

Iron activates microglia and directly stimulates indoleamine-2,3-dioxygenase activity in the N171-82Q mouse model of Huntington’s disease

PLOS ONE

Dear Dr. Fox,

Thank you for submitting your manuscript to PLOS ONE. After careful consideration, we feel that it has merit but does not fully meet PLOS ONE’s publication criteria as it currently stands. Therefore, we invite you to submit a revised version of the manuscript that addresses the points raised during the review process.

We look forward to receiving your revised manuscript.

Kind regards,

Jane Foster, PhD

Academic Editor

PLOS ONE

Journal Requirements:

Additional Editor Comments (if provided):

Reviewers' comments:

Reviewer's Responses to Questions

**Comments to the Author**

1. Is the manuscript technically sound, and do the data support the conclusions?

Reviewer #1: Partly

Reviewer #2: Yes

2. Has the statistical analysis been performed appropriately and rigorously? 

Reviewer #1: I Don't Know

Reviewer #2: Yes

3. Have the authors made all data underlying the findings in their manuscript fully available?

Reviewer #1: Yes

Reviewer #2: Yes

4. Is the manuscript presented in an intelligible fashion and written in standard English?

Reviewer #1: Yes

Reviewer #2: Yes

5. Review Comments to the Author

Reviewer #1: Overall the manuscript is clearly written with only a few typographical errors. Statistical analysis is described but it is beyond my expertise to evaluate the statistical methods employed. Methods are well described.

Major points:

1) One conclusion, that iron promotes neurodegeneration, is supported by brain weights and striatal volumes. Is there precedence in the literature that these measurements are sufficient to support this conclusion in the absence of any data specifically implicating neurons? Or would “brain and striatal atrophy” be a more appropriate description of what is being examined?

2) It is stated “NIS alone significantly increased Ido2 104 expression irrespective of mouse genotype (Fig. 1E,F)” but there are no indications of statistical significance in Figure 1E and 1F.

3) In the text EOC 20 cells should be referred to as “an immortalized microglial cell line” and not simply “microglia”

4) The statement in lines 195/196 about CCL2 activating microglia is inaccurate and needs to be revised.

Other points:

1) Please define KP at first use in abstract

2) Please state what deferoxamine does at first use in main text results.

3) In describing studies please be more consistent with use of terms- ironII, iron, ferrous iron and ferrous ammonium sulfate are all used at one place to describe in vitro studies

Reviewer #2: In this manuscript Donley and colleagues “utilized the neonatal iron supplementation model to investigate the relationship between iron, microglial activation and neurodegeneration in adult Huntington's Disease mice”. The authors showed that “neonatal iron supplementation in these mice promoted neurodegeneration and resulted in additional microglial activation”. The authors demonstrated that “that iron activates

IDO” and that “brain IDO activity is increased by HD”.

This is a dense manuscript from J. Fox’s laboratory that works for years in HD and iron. It is a well crafted study, that provides new insights in neurodegeneration and therefore deserves publication.

Statistical analyses, review of previous works and extensive materials and methods provided.

The authors did a wonderful effort to discuss their results and provide intelligent interpretation.

Figures are informative although a little too crowded.

Minor point

Results: Figure 1E,F, since NIS significantly increased Ido2 please include asterisks in the corresponding figure bars.

Potential Concerns:

None identified

No concerns were raised about dual publication, research ethics, or publication ethics. All animal permissions and IACUC protocols exist. At the designated endpoint, mice were humanely sacrificed using intra-peritoneal injections of B-euthanasia solution in the home cage.

All data are fully available without restriction and included in the manuscript.

Funding information from NIH is also provided.

6. PLOS authors have the option to publish the peer review history of their article (what does this mean?). If published, this will include your full peer review and any attached files.

Reviewer #1: No

Reviewer #2: No

---

## [Author Response · Author response to Decision Letter 0]

4 Apr 2021

All my comments are in the response to reviewer file and are pasted below.

Dear editor and reviewers: We appreciate the input on our manuscript. Responding to this has enabled us to make further improvements. Please see the specific comments below that are responses to all the reviews comments.

We hope that you will now find this manuscript acceptable for publication in PLOS ONE.

Sincerely, Jonathan Fox

Line numbers refer to the finalized version without track changes.

Reviewer 1: 

Major points:

1. The reviewer raises the question of whether loss of brain weight and striatal volume is sufficient to support the conclusion that there is neurodegeneration, or would brain/striatal atrophy be a more appropriate term to use. This a valid point. We do think there is significant precedence in the literature that our findings are sufficient to use the term “neurodegeneration”. There is overwhelming evidence in human HD that brain atrophy is driven to a large extent by neuron degeneration and loss (Curr Top Behav Neurosci. 2015;22:33-80). This may be also contributed by oligodendrocyte degeneration in white matter (PNAS May 7, 2019 116 (19) 9622-9627). The N171-82Q mouse model of HD demonstrate a number of changes that support neuron degeneration including TUNEL and silver stain positive neurons in striatum and or cortex (Curr Top Behav Neurosci. 2015;22:33-80; Neurobiology of Disease, June, 2008, 30;293-302). This is in addition to very well established loss of striatal and overall brain mass (Brain Pathol 2016 Nov;26(6):726-740). While we acknowledge that demonstrating neuron-specific changes would have enhanced the study, our interpretation of neurodegeneration is consistent with the current literature. To address this point in the text we have made the following addition. “These mice develop neurodegeneration as evidenced by loss of brain and striatal volume, and neuronal loss” and included a reference. Line 91.

2. Figs 1E and 1F have now been modified according to the reviewer’s suggestions. 

3. EOC 20 has now been changed to “an immortalized microglial cell line”. See lines 135 and 395.

4. Statement about CCL activating microglia – We agree with the reviewer that the original statement was inaccurate. We have now changed this to -”The chemokine CCL2 (MCP-1), expressed by both neurons and glia, recruits microglia and peripheral monocytes to sites of inflammation in the central nervous system. Expression of CCL2 is associated with microglial activation, but is an indicator of a more generalized, sustained inflammatory response”. See lines 207-210.

Minor points:

1. KP is now defined correctly in the abstract.

2. We have now explained what deferoxamine does. We changed the sentence to “IDO activity in HD mice that received NIS was significantly decreased by the potent iron (III) chelator deferoxamine added ex vivo in striata (p = 0.0067) and cortices (p = 0.0028)”. See lines 100-101.

3. We have now made the terminology around iron more consistent. We have removed the term ferrous and used “iron (II)” instead. Most of the time we do not designate the oxidation state of iron because it is not necessary, or the form of iron is unknown.

---

## [Editor Report · Decision Letter 1]

12 Apr 2021

Iron activates microglia and directly stimulates indoleamine-2,3-dioxygenase activity in the N171-82Q mouse model of Huntingtons disease

PONE-D-21-02271R1

Dear Dr. Fox,

We’re pleased to inform you that your manuscript has been judged scientifically suitable for publication and will be formally accepted for publication once it meets all outstanding technical requirements.

Kind regards,

Jane Foster, PhD

Academic Editor

PLOS ONE
---

## [Editor Report · Acceptance letter]

4 May 2021

PONE-D-21-02271R1 

Iron activates microglia and directly stimulates indoleamine-2,3-dioxygenase activity in the N171-82Q mouse model of Huntington’s disease 

Dear Dr. Fox:

I'm pleased to inform you that your manuscript has been deemed suitable for publication in PLOS ONE. Congratulations! Your manuscript is now with our production department. 

Kind regards, 

on behalf of

Dr. Jane Foster 

Academic Editor

PLOS ONE